



# Simulations of Spectral Polarimetric Variables measured in rain at W-band

Ioanna Tsikoudi[1,2], Alessandro Battaglia[3,4], Christine Unal[5,6], and Eleni Marinou[1]

[1]Institute for Astronomy, Astrophysics, Space Applications and Remote Sensing, National Observatory of Athens, Greece
[2]Department of Physics, National and Kapodistrian University of Athens, Greece
[3]Department of Environment, Land and Infrastructure Engineering , Politecnico of Torino, Turin, Italy
[4]Department of Physics and Astronomy, University of Leicester, Leicester, UK
[5]Geoscience and Remote Sensing, Delft University of Technology, Netherlands
[6]Climate Institute, Delft University of Technology, Netherlands

**Correspondence:** jtsik@noa.gr

**Abstract.** In this work, the T-matrix approach is exploited to produce simulations of spectral polarimetric variables (spectral differential reflectivity, $sZ_{DR}$, spectral differential scattering phase, $s\delta_{HV}$ and spectral differential correlation coefficient, $s\rho_{HV}$) for observations of rain acquired from a slant-looking W-band cloud radar. The spectral polarimetric variables are simulated with two different methodologies, taking into account the instrument noise and the stochastic movement of the raindrops introduced by raindrop oscillations and by turbulence. The simulated results are then compared with rain Doppler spectra observations from a W band millimeter-wavelength radar for moderate rain rate conditions. Two cases, differing in levels of turbulence, are considered. While the comparison of the simulations to the measurements presents a reasonable agreement for equi-volume diameters less than 2.25 mm, large discrepancies are found in the amplitude (but not the position) of the maxima and minima of $sZ_{DR}$ and, more mildly, of $s\delta_{HV}$. This pinpoints at a general weakness of the raindrops approximation with spheroids for simulating radar backscattering properties at W-band.

## 1 Introduction

Cloud radar observations are crucial for understanding cloud microphysics, as proposed in the groundwork laid by radar pioneers (Atlas et al., 1973; Lhermitte, 1990). In the last 25 years this is corroborated by an abundance of studies based on vertically-pointing spectral Doppler cloud radar observations in multi-frequency configurations and/or in synergy with lidar and radiometers for better characterising drizzle (e.g. O'Connor et al., 2005; Kollias et al., 2011; Luke and Kollias, 2013), rain (Kollias et al., 2001, 2002; Tridon et al., 2013; Tridon and Battaglia, 2015; Courtier et al., 2022), ice (Kalesse et al., 2016; Kneifel et al., 2016; Li et al., 2021; Luke et al., 2021), mixed-phase (Luke et al., 2010) and melting particles (e.g. Li and Moisseev, 2019; Mróz et al., 2021). Additionally, polarimetric variables are paramount for the characterization of the shape of hydrometeors, and are measured from all ground-based precipitation radar networks through sequence of scans at low elevation angles (Chandrasekar et al., 2023 and reference therein). Vertically pointing cloud radars miss most of the polarimetric information of hydrometeors (with the only exception of the linear depolarization ratio, (Mróz et al., 2021)), since



hydrometeors tend to fall with their maximum dimensions horizontally aligned. In order to overcome this limitation, more recently few sites started operating cloud radars with Doppler and polarimetric capabilities in slant observation mode (Spek et al., 2008; Myagkov et al., 2020; Mak and Unal, 2024; Unal and van den Brule, 2024). This configuration has the critical

advantage that particles with different sizes are separated in the spectral domain (because they have different sedimentation velocities), which allows to disentangle the contributions of different particle types. Especially when increasing the frequency (i.e. in presence of multiple resonances across the range of the particle size distribution (PSD)), the polarimetric variables that result from the integration over the entire PSD tend to average out the characteristic features (often both positive and negative) of the single scattering polarimetric properties. Specifically, for Ka and W-band observations of rain at 45 deg. elevation angle,

Unal and van den Brule (2024) have demonstrated that, by using the Rayleigh plateau as previously proposed in literature (Tridon et al., 2013; Myagkov et al., 2020), it is possible to separate the propagation and backscatter contributions in the spectral domain for the polarimetric variables (namely the differential phase shift and the differential attenuation). Then the differential phase at backscattering can be used to infer the characteristic droplet diameter of the droplet size distribution (DSD). Incidentally W-band polarimetric radar observations at slant angles have been also proposed in the framework of the

ESA spaceborne WIVERN mission (Illingworth et al., 2018; Battaglia et al., 2022), which aims to measure in-cloud winds by using the polarization diversity technique with an antenna conically scanning at an incidence angle of $41.6°$. Although in WIVERN case no spectral measurements are envisaged, this mission will provide an unprecedented abundance of incidentally cloud radar polarimetric observations globally.

Spectral polarimetric observations, utilizing either slant or horizontal profiling, effectively distinguish hydrometeors from

clutter (Bachmann and Zrnić, 2007; Moisseev and Chandrasekar, 2009; Unal, 2009; Chen et al., 2022) and also enable the characterization of various hydrometeors (Spek et al., 2008; Pfitzenmaier et al., 2018; Wang et al., 2019; Lakshmi et al., 2024). In the case of rain, Moisseev et al. (2006) derived the shape-size relationship, while Yanovsky (2011) explored the effects of turbulence on spectral $Z_{dr}$. These studies were conducted at centimeter-wavelength frequencies.

In order to build quantitative retrieval algorithms based on spectral polarimetric observations, forward model simulators of

the spectral polarimetric spectra themselves are needed. Simulations of Doppler spectra observed by ground-based vertically pointing radars have been pioneered by Zrnić (1975) and has been applied to different hydrometeors and to millimeter radars by different authors (e.g. Kollias et al., 2011; Tridon and Battaglia, 2015; Courtier et al., 2024), also including turbulence effects and raindrop inertia (Zhu et al., 2023). The simulation of spectral polarimetric spectra (Myagkov et al., 2020; Unal and van den Brule, 2024) has been explored only marginally because slant observations are not so common.

Goal of this study is to describe the methodology for simulating polarimetric spectral variables including white and stochastic noise of a real radar spectrum, as well as the impact of atmospheric turbulence and to compare simulations with observed spectra for rain observations. Rain electromagnetic scattering properties have been historically computed by assuming spheroidal or Chebyshev shapes (both rotationally symmetric) via the T-matrix method (Mishchenko et al., 2000). Such models have been found satisfactory to explain radar and radiometeric measurements in the S, C, X, Ku and Ka band (Battaglia et al., 2010;

Kumjian et al., 2019; Teng et al., 2018) but they have also been used to simulate higher radar frequencies (Aydin and Lure, 1991; Kneifel et al., 2020; Unal and van den Brule, 2024). However, raindrops generally change due to oscillations, which





cause departure from rotationally symmetric shape, and make T-matrix tools impractical since they hinge upon the assumption of rotationally symmetric particles. Different studies have highlighted the strong impact of the shape assumptions in modifying the polarimetric variables (e.g. Ekelund et al., 2020 compared sphere, spheroids and equilibrium/Chebyshev drops) particu-
larly when considering particles in the resonance regions (Thurai et al., 2007) (that occur in the 5.5–7-mm-diameter region at C band and at smaller sizes and in multiple ranges when increasing frequency). Such studies however are based on study of the DSD-integrated polarimetric variables and therefore do not fully capture the impact of the shape on each single particle. Combining Doppler and polarimetric measurements spectral polarimetry has the potential to test hydrometeor shape models and their associated scattering properties in great detail.

The paper is structured as following. First we detail the methodology for simulating the cloud radar spectra and polarimetric variables (Sect. 2); then we present the results of our simulations, describe the observational dataset, compare simulations and observations, and discuss the implications of our findings.

## 2   Methodology for simulations

### 2.1   Rain scattering properties simulated by T-matrix

The simulations are generated by using a Python package for computing the electromagnetic scattering properties of nonspherical particles using the T-matrix method (Leinonen, 2014).

In this study, the rain scattering properties are exclusively targeted. The backscattering amplitude matrix, $S$, and the phase matrix, $Z$, (Chapt. 16 in Mishchenko et al., 2000) are calculated for drops of different diameters D, with axis ratios parameterized according to Keenan et al. (2001); Andsager et al. (1999); Beard and Chuang (1987); Bringi and Chandrasekar (2001), as
demonstrated in Eq. (1).

$$\frac{a}{b}(D) = \begin{cases} 1/(0.9939 + 0.00736 \cdot D - 0.018485 \cdot D^2 + 0.001456 \cdot D^3) & D < 0.89\,mm \\ 1/(1.0048 + 5.7 \cdot 10^{-4}D - 2.628 \cdot 10^{-2}D^2 + 3.682 \cdot 10^{-3}D^3 - 1.677 \cdot 10^{-4}D^4) & D \geq 0.89\,mm \end{cases} \quad (1)$$

where a/b denotes the ratio of the major to minor axis of the oblate spheroid.

The brown line in Figure 1 represents the axis ratio parametrization used in this study, and is plotted against the equivalent relationship of Thurai et al. (2008) (green dashed line) and the axis ratio of spheres (purple dotted line). The first two lines
present great agreement for particles with equi-volume diameter up to 3 mm. Very small droplets are conceived as perfect spheres (axis ratio ≈ 1). As their size increases, drops are modelled as spheroidal particles and an oblate shape is assumed (axis ratio > 1). The scattering geometry of the simulation corresponds to a radar pointing at a 45° elevation angle. Raindrops are assumed to be partially aligned with their maximum dimension preferentially on the horizontal plane: scattering properties are averaged over Gaussian distributions of canting angles with different standard deviations. The raindrops are assumed to be
at 10°C and the refractive index of water at this temperature is $3.2 - 1.8j$ at 94 GHz (Lhermitte, 1990).



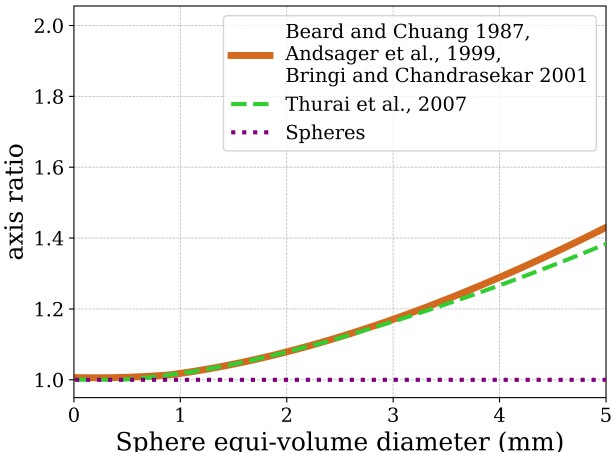

**Figure 1.** Axis ratio (major to minor axis) parametrization as a function of equi-volume diameters. The brown line is used in this study and is calculated according to Keenan et al. (2001); Andsager et al. (1999); Beard and Chuang (1987); Bringi and Chandrasekar (2001). The green dashed line is the parametrization of Thurai et al. (2008) and the purple dotted line is the axis ratio of spheres.

### 2.1.1 Computation of single-particle Polarimetric Variables

The phase matrix $Z$ describes how an electromagnetic wave is scattered by a particle and how the scattering affects its polarization state (Mishchenko et al., 2000). It is a 4x4 matrix that transforms the Stokes vector of an incident electromagnetic wave into the Stokes vector of the scattered wave. From the elements $Z_{ij}$(D) of this matrix the following backscattering quantities can be computed:

– Backscattering cross sections for V-polarized and H-polarized radiation:

$$\sigma_{VV}(D) = 2\pi(Z_{11} + Z_{12} + Z_{21} + Z_{22}) \qquad [\text{mm}^2]$$
$$\sigma_{HH}(D) = 2\pi(Z_{11} - Z_{12} - Z_{21} + Z_{22}) \qquad [\text{mm}^2]$$

(2)

– Differential reflectivity

$$Z_{DR}(D) = 10\log_{10}\frac{\sigma_{HH}(D)}{\sigma_{VV}(D)} \qquad [\text{dB}]$$

(3)

– Copolar correlation coefficient

$$\rho_{HV}(D) = \frac{\sqrt{(Z_{33} + Z_{44})^2 + (Z_{43} - Z_{34})^2}}{\sqrt{(Z_{11} - Z_{12} - Z_{21} + Z_{22})(Z_{11} + Z_{12} + Z_{21} + Z_{22})}}$$

(4)

– Differential Phase

$$\delta_{HV}(D) = \arctan\left(\frac{Z_{43} - Z_{34}}{Z_{33} + Z_{44}}\right) \qquad [\text{degrees}]$$

(5)





The normalised backscattering cross section of an oblate spheroidal raindrop is shown in Fig. 2 with brown color. The
axis ratio for this computation is the same as the brown line of Fig. 1. The dashed green line represent the same quantity,
but computed by using the Thurai et al. (2008) axis ratio parametrization (green line in Fig. 1). The same applies for the
purple dotted line which is produced by using the spheres' axis ratio. The two different spheroids parametrizations result in
nearly identical curves, indicating that the choice of axis ratio for oblate shapes does not significantly affect the backscattering
cross section behavior. In contrast, the spherical parametrization shifts the Mie notches slightly to the left, due to the different
geometry of the scatterers. The positions of the first, second, and third Mie notches, are indicated by the blue dashed lines at
$D = 1.68$ mm, $D = 2.88$ mm, and $D = 4.13$ mm, respectively.

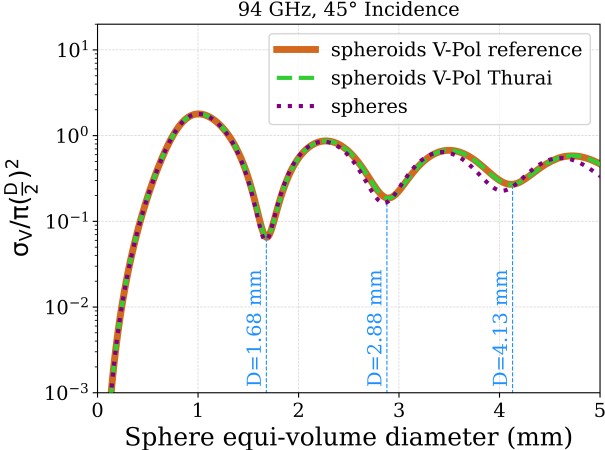

**Figure 2.** Normalised backscattering cross section of oblate spheroidal model raindrops when pointing at $45°$ elevation as function of the
equi-volume spherical drop diameter $D$. The light blue dashed lines indicate the first ($D = 1.68$ mm), second ($D = 2.88$ mm), and third
($D = 4.13$ mm) Mie notches.

Some T-matrix results for the polarimetric variables are displayed in Fig. 3 and 4: different drop orientation conditions
and raindrops axial ratios are considered. The black dashed and the blue lines are calculated by assuming perfectly oriented
raindrops with axis ratio parametrization as proposed by Thurai et al. (2008) and according to Eq. 1, respectively. Those two
lines are almost identical up to approximately 3 mm diameters but they diverge afterwards. Notably, for larger raindrops,
the black dashed line aligns closely with the light blue line, which represents a wobbling raindrop with a $5°$ canting angle
on average. This suggests that the same amplitude of the maxima and minima in the spectral polarimetric variables can be
achieved by different combinations of axis ratio parameterizations and varying degrees of wobbling. Therefore in the following
the parametrization of Eq. 1 is used in combination with different degrees of wobbling.
The differential phase ($\delta_{HV}$) refers to the phase shift introduced at backscattering between the horizontally and vertically
polarized components of the received radar signal, providing information about the shape and orientation of hydrometeors.
In Fig. 3 (right), very small diameters are considered as spheres and present $\delta_{HV} = 0$, indicating the same phase shift for



both horizontal and vertical polarizations at backscattering. As the diameter increases, $\delta_{HV}$ starts to show fluctuations. The canting angles introduce variability in the orientation of the drops within the radar beam, leading to variations in the observed $\delta_{HV}$. Larger diameters exhibit more pronounced fluctuations due to the combined effects of resonance and the transition from spherical to oblate shapes. When particles are randomly oriented (red line in Fig. 3), their orientations are distributed uniformly in all directions. In this case, the ensemble-averaged response over all possible orientations lead to cancellation effects in the differential phase ($\delta_{HV}$=0, Fig. 3-right) and in the differential reflectivity ($Z_{DR}$, Fig. 4-left). The cancellation occurs because, for a medium which is a mixture of randomly oriented particles, the off-diagonal elements $Z_{12}, Z_{21}, Z_{34}, Z_{43}$ of the backscattering matrix become zero (as shown in Mishchenko et al. (2000), Chapter 3, Table II), thus leading to zero $Z_{DR}$ and $\delta_{HV}$ (see Eq. (3)-(5)). The blue dashed lines of Fig. 3 and 4 indicate the positions of the Mie notches, as depicted in Fig. 2. The first two minima of $\delta_{HV}$ coincide with the Mie notches, while the $Z_{DR}$ is approximately zero at these points. Moreover, the diameters of the minima ($D_1, D_3, D_5$), and maxima ($D_2, D_4, D_6$), are demonstrated for $Z_{DR}$.

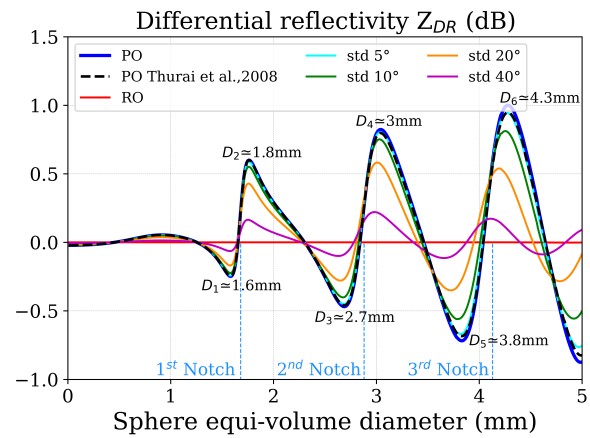
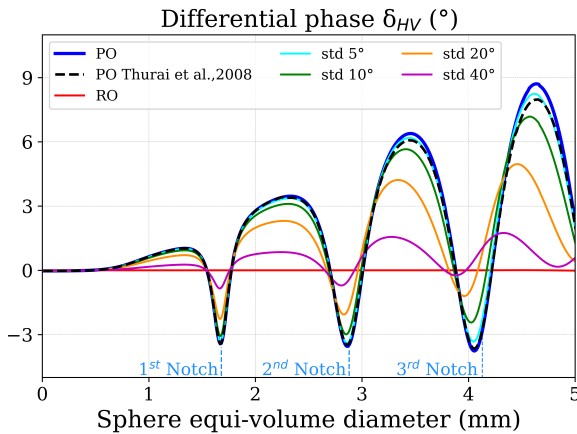

**Figure 3.** Simulations of differential reflectivity $Z_{DR}$ (left) and differential phase $\delta_{HV}$ (right) as a function of sphere equivalent-volume diameters, for a 94 GHz radar pointing at 45°. PO (Perfect Orientation) and RO (Random Orientation) are represented by the dark blue and red lines, respectively, derived with axis ratio parametrization according to Eq. (1). The black dashed line corresponds also to perfectly oriented raindrops with axis ratio parameterization as proposed by Thurai et al. (2008). The remaining lines represent different degrees of raindrop wobbling, with a Gaussian distribution around the horizontal with standard deviations of 5°(light blue), 10°(green), 20°(orange), and 40°(pink).

The copolar correlation coefficient ($\rho_{HV}$) quantifies the correlation between the horizontally and vertically polarized components of the radar signal. In Fig. 4, perfectly oriented drops (blue and black dashed lines) have $\rho_{HV}$=1. On the other hand, rain drops with orientation or tilt of the drop axis relatively to the direction of motion (canting) have $\rho_{HV}$ slightly lower than 1 showing a minimum loss of correlation between the two different polarization states. Even when considering randomly oriented raindrops $\rho_{HV}$ never falls short of 0.986. Realistic values of canting generally do not exceed 10° (Mishchenko et al., 2000).





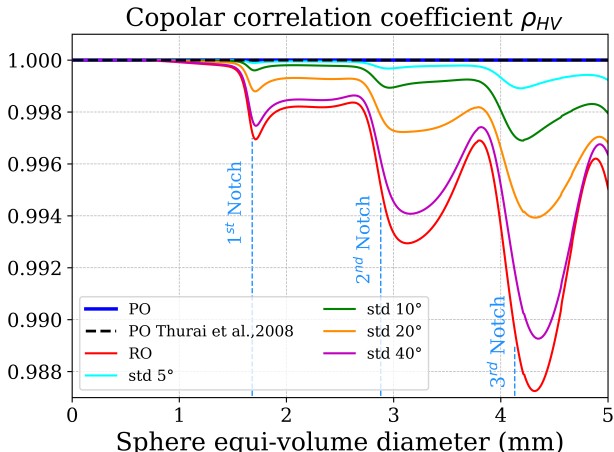

**Figure 4.** Same as in Fig. 3 for the copolar correlation coefficient $\rho_{HV}$ as a function of sphere equivalent-volume diameters, for a 94 GHz radar pointing at an elevation of $45°$.

### 2.1.2 Drop Size Distribution and Rainfall Velocities

The gamma distribution is a mathematical shape typically used to represent the variability of a natural rainfall Drop Size Distribution (DSD) (Ulbrich, 1983):

$$N(D) = N_0 D^\mu exp(-\Lambda D) \qquad [mm^{-1}m^{-3}] \tag{6}$$

D [mm] is the equivalent spherical drop diameter, $\mu$ is the dimensionless shape parameter, $N_0$ $[mm^{-1-\mu}m^{-3}]$ is the number concentration parameter and $\Lambda$ [$mm^{-1}$] is the slope parameter. The three parameters ($N_0$, $\mu$, and $\Lambda$) of the gamma distribution enable a wide range of rainfall situations to be described. $\Lambda$ can be derived from $\Lambda = \frac{4+\mu}{D_m}$, where $D_m$ [mm] is the mass-weighted mean diameter (Ulbrich and Atlas, 2007; Testud et al., 2001).

Important for Doppler applications, the larger the drops, the faster the terminal fall speed, $v_T$. The relationship between the drop diameters and the corresponding velocities is parameterized in SI units as:

$$v_T(D) = \begin{cases} v_{cloud} = 1.2 \cdot 10^8 \cdot \left(\frac{D}{2}\right)^2 & D < 0.11\,mm \\ v_{drizzle} = 8333 \cdot \frac{D}{2} - 0.0833 & 0.11 < D < 0.86\,mm \\ v_{rain} = 9.65 - 10.3 \cdot e^{-0.6 \cdot 10^3 \cdot D} & D > 0.86\,mm \end{cases} \tag{7}$$

A factor $\left(\frac{\rho_0}{\rho}\right)^{0.4}$ with $\rho_0$ being the density at sea level applies for different air densities.

In Fig. 5, raindrop terminal velocities are plotted against the diameters according to Eq. (7) and the parametrization from Thurai and Bringi (2005) (brown and dashed black line, respectively). The relative difference between the two velocity parametrizations never exceeds 2%. Therefore, when mapping terminal velocities into diameters, this translates into simi-





lar relative uncertainties in the determination of diameters for any given velocity. For instance the position of the first (second) Mie notch is expected to occur at terminal velocities of $5.89 \pm 0.11\,m/s$ ($7.82 \pm 0.15\,m/s$).

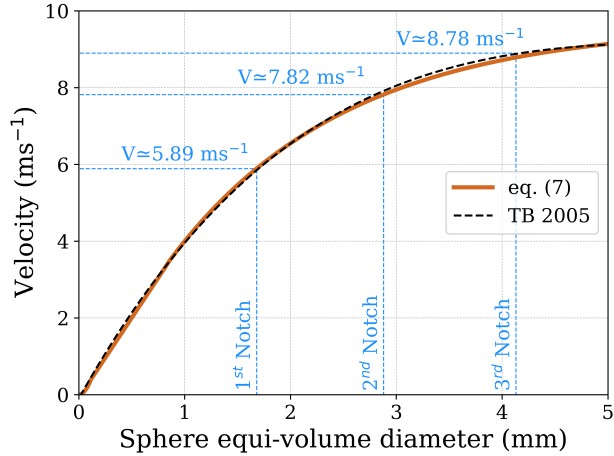

**Figure 5.** Terminal fall speed $v_T(D)$ as a function of the equi-volume spherical drop diameter D, for equation (7) with brown thick line, and for Thurai and Bringi (2005) with dashed black line.

## 2.2   Simulation of spectral polarimetric variables

Two methodologies for simulating spectral polarimetric variables as observed from a W-band cloud radar will be presented in this paper. The first was developed based on (Yu et al., 2012; Zrnić, 1975), while the second on (Thurai et al., 2008) and
(Chandrasekar, 1986). Notably, both methods show very good agreement and will be described in detail in Secs. 2.2.1, 2.2.2. Some preliminary processing is needed for both methodologies as discussed below.

Firstly an ideal co-polar spectrum $S_{VV}$ for the V-channel is independently generated for each diameter:

$$S_{VV}(v_{LoS}) = \frac{\lambda^4}{\pi^5 |K|^2} \, N(D) \, \sigma_{VV}(D) \, \frac{1}{\sin\theta_{el}} \, \frac{dD}{dv_T(D)} \tag{8}$$

where $\lambda$ is the radar wavelength, $|K^2|$ is derived from the dielectric factor of water, $N(D)$ is the DSD (see Sec. 2.1.2), $\sigma_{VV}$
is the backscattering cross section for V channel (Sect. 2.1.1) and $v_{LoS}(D) = \sin\theta_{el}v_T(D) + w_{LoS}$ denotes the Line of Sight (LoS) Doppler velocities of the drops at the given elevation angle $\theta_{el}$. $v_{LoS}$ is the sum of the component of the raindrop terminal velocity and of the wind speed along the LoS. The spectrum is mapped into the velocity domain via Eq. (7) and sampled in correspondence to the velocity points $v_j$ with $j = 1, 2, \dots, N_{FFT}$ where $N_{FFT}$ is the number of FFT points as dictated by the Doppler velocity resolution and Nyquist interval envisaged for any given radar system. The samples are indicated with
$S_{VV}(v_j)$. Similarly, H-channel spectrum can also be produced at each velocity bin by replacing $\sigma_{VV}(D)$ with $\sigma_{HH}(D)$ in Eq. (8).





The cross spectrum, denoted as $S_{HV}(D)$, is derived as follows:

$$S_{HV}(v_{LoS}) = \frac{\lambda^4}{\pi^5 |K|^2} \, N(D) \, \sqrt{\sigma_{VV}(D)\sigma_{HH}(D)} \, \frac{1}{\sin\theta_{el}} \, \frac{dD}{dv_T(D)} \, \rho_{HV}(D) \, e^{\imath\delta_{HV}(D)} \tag{9}$$

where $\imath = \sqrt{-1}$, $\rho_{HV}(D)$ is the correlation coefficient between the V and H channels, and $\delta_{HV}(D)$ is the phase difference
between the V and H channel signals, as described in Eqs. (4-5). The spectrum is sampled similarly to the $V$ channel spectrum
at velocity points $v_j$ with $j = 1, 2, \ldots, N_{FFT}$, and the samples are denoted as $S_{HV}(v_j)$. Note that each Doppler velocity
spectrum can be converted into the frequency domain by using the relationship $f_D = \frac{2v_{LoS}}{\lambda}$ between frequency Doppler shift,
$f_D$, and $v_{LoS}$.

Generally spectra are derived at any given range via FFT of the time series of radar sampled voltage signals, the so called $I$
(in-phase) and $Q$ (quadrature) signals collected at the same range distance (Doviak and Zrnić, 1993). In the following complex
voltages will be identified with calligraphic style letters (e.g. $\mathcal{V}, \mathcal{N}$). Also, such voltages will be always expressed in the velocity
domain as indicated by their functional argument. They correspond to the FFT of the voltages expressed in the time domain.

### 2.2.1 Methodology I: direct computation of $I$ and $Q$s in the frequency domain

This method allows Doppler spectra to be simulated by working only in the velocity (frequency) domain. Following Yu et al.
(2012) the time series of complex voltage signal in the V channel in the velocity domain can be written as:

$$\mathcal{V}_V^{[1]}(v_j, k) = \sqrt{-S_{VV}(v_j)\ln u_{jk}^{[1]}} \, e^{\imath\theta_{jk}^{[1]}} \qquad \begin{matrix} j=1,2,...,N_{FFT} \\ k=1,2,...,K \end{matrix} \tag{10}$$

where $u^{[1]}$ and $\theta^{[1]}$ are independent, identically distributed random variables with uniform distribution between 0 and 1 and be-
tween $-\pi$ and $\pi$, respectively. This process can be repeated $k = 1, 2, \ldots, K$ times, in order to generate $K$ independent stochastic
realizations of the same spectrum. Similarly, for the H channel in the velocity domain:

$$\mathcal{V}_H(v_j, k) = \sqrt{sZ_{DR}(v_j)}[s\rho_{HV}(v_j) \, \mathcal{V}_V^{[1]}(v_j, k)$$
$$+ \sqrt{1 - s\rho_{HV}{}^2(v_j)} \, \mathcal{V}_V^{[2]}(v_j, k)] \, e^{\imath \, s\delta_{HV}(v_j)} \qquad \begin{matrix} j=1,2,...,N_{FFT} \\ k=1,2,...,K \end{matrix} \tag{11}$$

where the spectral variables $s\rho_{HV}$, $s\delta_{HV}$ and $sZ_{DR}$ are generated as described in Sect. 2.1 for each velocity bin $j$, but also hold
the prefix $s$ in the notation to differentiate them from the commonly used integral polarimetric variables. $\mathcal{V}_V^{[2]}(v_j, k)$ is generated
according to (10) with the same model spectrum $S_{VV}(v)$, but with a second independent sequence of random numbers ($u^{[2]}$
and $\theta^{[2]}$). This process is repeated for each velocity bin for a total of $N_{FFT}$ spectral points within the Nyquist interval. The
inverse Fourier transform of $\mathcal{V}_V(v_j)$ and $\mathcal{V}_H(v_j)$ with $j = 1, 2, \ldots, N_{FFT}$ represent simulated time series of complex signals
for the V and H channels. For the implementation of white noise, an approach similar to Eq. (10) is used:

$$\mathcal{N}_V(v_j, k) = \sqrt{-N_V \ln u_{jk^{[3]}}} \, e^{\imath\theta_{jk}^{[3]}}$$
$$\mathcal{N}_H(v_j, k) = \sqrt{-N_H \ln u_{jk}^{[4]}} \, e^{\imath\theta_{jk}^{[4]}} \qquad \begin{matrix} j=1,2,...,N_{FFT} \\ k=1,2,...,K \end{matrix} \tag{12}$$

where $N_V$ and $N_H$ are the noise power levels for the V and H channel corresponding to the prescribed values of signal-to-noise
ratios ($SNR$), and $u^{[3]}$, $\theta^{[3]}$, $u^{[4]}$, $\theta^{[4]}$ are again generated independently.





The complex numbers that represent the simulation of the noisy I and Qs in the frequency domain for the V and H channels are calculated from:

$$\mathcal{S}_V(v_j, k) = \mathcal{V}_V(v_j, k) + \mathcal{N}_V(v_j, k)$$
$$\mathcal{S}_H(v_j, k) = \mathcal{V}_H(v_j, k) + \mathcal{N}_H(v_j, k)$$

$$j = 1, 2, ..., N_{\text{FFT}}$$
$$k = 1, 2, ..., K$$

(13)

### 2.2.2    Methodology II: correlation matrix

Alternatively the $I$ and $Q$ generation can be performed using the methodology proposed by Unal and Moisseev (2004) based
on the correlation matrix. First the correlation matrix $R$ is built with the Doppler power spectra in the diagonal terms and the cross-polar spectrum in the antidiagonal elements as:

$$R(v_j) = \begin{pmatrix} S_{VV}(v_j) + N_V & S_{HV}(v_j) \\ S_{HV}^\star(v_j) & S_{HH}(v_j) + N_V \end{pmatrix} \qquad j = 1, 2, \ldots N_{FFT}$$

(14)

with all terms given by Eqs. (8-9). Noise has also been included but with no copolar correlation. Because $R$ is Hermitian and positive definite, it may be written as $R = T^\dagger T$ via Cholesky decomposition, where $^\dagger$ denotes Hermitian transpose. Given
$2N_{FFT}$ zero-mean independent standard circular Gaussian random variables, $y_1, y_2, \ldots y_{2N_{FFT}}$ [i.e. $y_j = 1/\sqrt{2}(\xi_j + \imath\, \eta_j)$ where $\xi_j$ and $\eta_j$ are normally distributed with mean equal 0 and standard deviation equal 1], the complex numbers

$$\begin{bmatrix} \mathcal{S}_V(v_1) \\ \mathcal{S}_H(v_1) \\ \mathcal{S}_V(v_2) \\ \mathcal{S}_H(v_2) \\ \vdots \\ \mathcal{S}_V(v_{FFT}) \\ \mathcal{S}_H(v_{FFT}) \end{bmatrix} = T^\dagger \begin{bmatrix} y_1 \\ y_2 \\ y_3 \\ y_4 \\ \vdots \\ y_{2N_{FFT}-1} \\ y_{2N_{FFT}} \end{bmatrix}$$

(15)

have components distributed as normally distributed variables with zero mean and with correlation provided by $R$. The procedure can be repeated $K$-times to simulate $K$ different spectra.

### 2.2.3    Computation of polarimetric variables from $I$ and $Q$s

Once the $I$ and $Q$s are obtained with either of the two methodologies, then noisy Doppler spectra can be computed as a spectral average of $K$ spectra as:

$$S_{VV}(v_j) \;=\; \langle |\mathcal{S}_V(v_j)|^2 \rangle = \frac{1}{K} \sum_{k=1}^{K} |\mathcal{S}_V(v_j, k)|^2$$

(16)

$$S_{HH}(v_j) \;=\; \langle |\mathcal{S}_H(v_j)|^2 \rangle = \frac{1}{K} \sum_{k=1}^{K} |\mathcal{S}_V(v_j, k)|^2$$

(17)



The spectral polarimetric variables $s\rho_{HV}(v)$ and $s\delta_{HV}(v)$ are calculated according to Mishchenko et al. (2000):

$$s\rho_{HV}(v_j)e^{\imath s\delta_{HV}(v_j)} = \frac{\langle \mathcal{S}_H(v_j)\mathcal{S}_V^{\star}(v_j)\rangle}{\sqrt{\langle |\mathcal{S}_H(v_j)|^2\rangle \langle |\mathcal{S}_V(v_j)|^2\rangle}} \tag{18}$$

where $\langle \mathcal{S}_H(v_j)\mathcal{S}_V^{\star}(v_j)\rangle$ is the average $\frac{1}{K}\sum\limits_{k=1}^{K}\mathcal{S}_H(v_j,k)\mathcal{S}_V^{\star}(v_j,k)$.

### 2.2.4   Inclusion of turbulence in the simulations

Understanding the effects of turbulence on the Doppler spectrum is crucial for improving the accuracy of radar observations

and their interpretation. Atmospheric turbulence causes random fluctuations in the velocity of hydrometeors, thus broadening

the Doppler spectrum. All droplets are here assumed to have no inertial effects and therefore act like perfect tracers. Thus,

to introduce the turbulent motions of drops in the simulations, the Doppler spectra must be convolved with a turbulence term

$S_{air}$:

$$S_{VV}^{turb}(v_{LoS}) = (S_{VV} * S_{\mathrm{air}})(v_{LoS}) = \int\limits_{-\infty}^{\infty} S_{VV}(v_{LoS} - \xi)\, S_{\mathrm{air}}(\xi)\, d\xi \tag{19}$$

where $\xi$ is the convolution variable and $S_{air}$ accounts for the turbulent motions within the atmosphere:

$$S_{air}(v) = \frac{1}{\sqrt{2\pi}\sigma_t}\, e^{-\frac{v^2}{2\sigma_t^2}} \tag{20}$$

with $\sigma_t$ expressing the turbulence broadening of the Doppler spectrum. Equations similar to Eq. (19) can be used to compute

the turbulence-broadened spectra $S_{HH}^{turb}(v)$ for H-polarized radiation, as well as for $S_{HV}^{turb}(v)$. Then the broadened $sZ_{DR}^{turb}(v)$

can be computed as the ratio of $S_{HH}^{turb}(v)$ to $S_{VV}^{turb}(v)$ whereas the turbulent-broadened parameters $s\rho_{HV}^{turb}$ and $s\delta_{HV}^{turb}$ are then

calculated respectively as the amplitude and the phase of the variable:

$$s\rho_{HV}^{turb}(v)e^{\imath s\delta_{HV}^{turb}(v)} = \frac{S_{HV}^{turb}(v)}{\sqrt{S_{HH}^{turb}(v)S_{VV}^{turb}(v)}} \tag{21}$$

For the generation of the $I$ and $Q$s:

   – for methodology 1 (Sect. 2.2.1) the simulated spectral polarimetric variables $sZ_{DR}^{turb}(v)$, $s\delta_{HV}^{turb}(v)$, $s\rho_{HV}^{turb}(v)$ will replace
     the ideal quantities in Eq. (11);

– for methodology 2 (Sect. 2.2.2) $S_{VV}^{turb}$, $S_{HH}^{turb}$, and $S_{HV}^{turb}$ are directly used in the definition of the correlation matrix in
     Eq. (14).

## 3   Comparisons with measurements

To assess the accuracy of the cloud radar simulation methods, we compare the measurements and the simulated data. This

comparison aims to validate the performance of the simulations and identify any discrepancies that may arise from model





assumptions or parameter settings. The cloud radar measurements were obtained using a RPG Frequency Modulated Continuous Wave (FMCW) Dual Polarization W-band Cloud Doppler Radar, operating at 94 GHz. The radar system was configured to investigate polarimetric and spectral polarimetric measurements of clouds and precipitation in the troposphere during four months (January-April 2021). The models described in 2.2.1, 2.2.2 were initialized based on the characteristics (SNR, PRF, FFT bins) of the real measurements to generate simulated radar data for the comparison with the real data.

Two case studies from 3 February 2021 are presented, both characterized by moderate rainfall, with rain rates approximately between 6 and 7 mm/h. The first one focuses on a spectrum acquired at an altitude of 105 meters above ground level, while the second one targets a spectrum at 484 meters. The cases differ primarily in the level of atmospheric turbulence observed at specific heights. Higher altitudes are usually characterized by significantly less turbulence relatively to lower levels, because turbulence is often generated by surface heating and friction. The measured spectrogram on the vertical channel $S_V$, and

the polarimetric variables $sZ_{DR}$, $s\delta_{HV}$ and $s\rho_{HV}$, are presented in Fig. 6. The x-axis represents the Doppler velocity, $v_{LoS}$, corresponding to the unfolded measured Doppler velocity. The spectral signatures associated with small raindrops appear on the left side of the spectra. As raindrop sizes increase and become comparable to the radar wavelength, non-Rayleigh scattering occurs, leading to resonance features observed on the right side of the spectra.

To facilitate the comparison between simulations and observational data, the terminal velocity, $v_T$, was selected for the

velocity axis in Sect. 3.1 and 3.2. Accordingly, the Doppler velocities shown in Fig. 6 were first adjusted along the velocity axis to remove the contribution of the radial wind, $w_{LoS}$. This correction was achieved by identifying the first Mie scattering minimum (Kollias et al., 2002). At an elevation angle of $\theta_l=45°$, the first Mie minimum corresponds to a velocity of $5.89 \sin\theta_l=$ 4.16 m$s^{-1}$. The resulting corrected Doppler velocities, $v_{LoS}$-$w_{LoS}$, were then divided by $\sin\theta_l$, yielding an estimate of the terminal velocities for the observations.

A comparison between measured and simulated $s\rho_{HV}$ is challenging. The measurement of $s\rho_{HV}$ is subjected to biases (particularly at low signal-to-noise levels, (Touzi et al., 1999)) and is affected by radar-specific characteristics (e.g. antenna-related) which are difficult to be quantified and accounted for (Myagkov et al., 2024). Therefore $s\rho_{HV}$ is not further analyzed in this paper.

### 3.1   Case Study 1: More turbulent conditions

The Doppler spectrum measured at a height of 105 m is presented in Figure 7 with a black line. The presence of turbulence is depicted on the broadening effect of the spectrum and the notches are smoothed out. To accurately match the measured radar spectrum, a variety of gamma Drop Size Distributions (DSDs) were produced by adjusting the parameters described in 2.1.2, aiming to find the DSD that best fit the observed spectrum (blue line). Different combinations of $\mu$, $N_0$, $D_m$ (from eq. 6) and $\sigma_t$ (from eq. 20) are tested to better represent the real measurement. To identify the optimal fit, the Least Squares Method was

employed. This method minimizes the sum of the squared differences between the measured and simulated spectra, ensuring that the best-fitting gamma DSD is selected. The spectra are compared in logarithmic scale rather than in linear units to better capture the wide dynamic range of radar reflectivity. This way, both high and low reflectivity values are appropriately weighted, avoiding dominance by large values that occur in linear comparisons. In order to avoid overfitting the tails of the spectrum (and





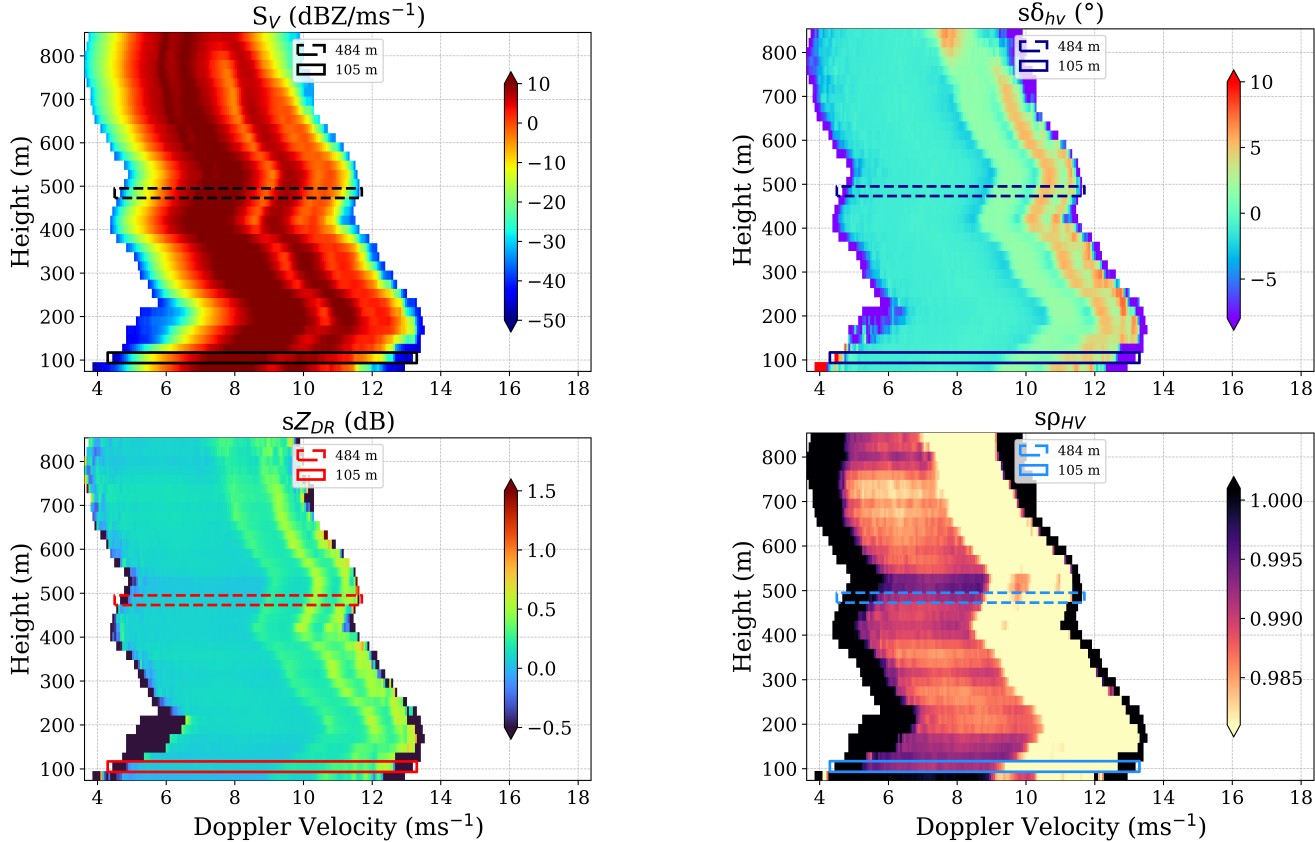

**Figure 6.** Event of 03 Feb 2021, 12:40 UTC with vertical profiles for reflectivity (top left), differential reflectivity (bottom left), differential phase shift (top right), correlation coefficient (bottom right) spectra. The two levels that are used for case studies are marked by the solid (105 m) and dashed (484 m) rectangles.

deteriorating the fits of the high SNR part of the spectrum, e.g. in correspondence to the Mie notch), only the part of the
spectrum above the purple dashed line at -8 $dBZ/ms^{-1}$ is fitted. This threshold is an empirical rule of thumb derived from this study, which primarily focused on cases with rain rates between 5-9 mm/hr.

In Figure 8, the black lines represent the measured spectral polarimetric variables $sZ_{DR}$ (left) and $s\delta_{HV}$ (right), while the blue and red lines are the results of the two simulation methods. In order to provide a consistent reference for spherical raindrops the measured $sZ_{DR}$ and $s\delta_{HV}$ were adjusted along the y-axis to 0 dB and 0°, respectively. This correction accounts
for propagation effects and instrument miscalibrations of the polarimetric variables.

There is excellent agreement (within the stochastic noisiness) between the two methods used for generating the simulations (blue and red lines) for the two variables. This gives confidence in the methodologies described in Sect. 2.2. On the other hand while the comparison between cloud radar simulations and measurements exhibits some correlation, there are notable



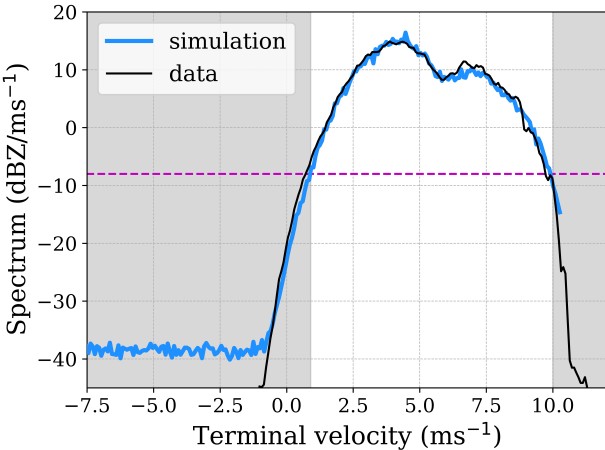

**Figure 7.** 03 February 2021, 12:40 UTC, 105m, Measured Doppler Spectrum (black line) and optimum-fitted Gamma DSD (blue line). The purple dashed line indicates the threshold for applying the Least Squares Method in order to find the optimum fit. The parameters that characterize the fitted spectrum are $\mu = 0$, $D_m = 1.8\, mm$, $N_0 = 987\, mm^{-1-\mu}\, m^{-3}$, and $\sigma_t = 0.5\, ms^{-1}$

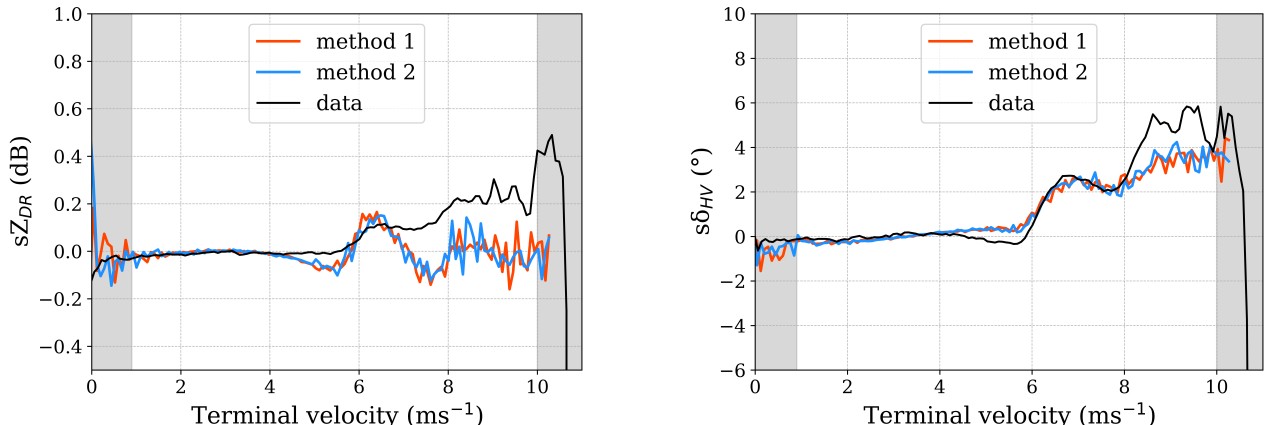

**Figure 8.** Spectral polarimetric variables of case study 1. Left panel: spectral differential reflectivity $sZ_{DR}$, Right panel: spectral dfferential phase $s\delta_{HV}$. The black lines represent the measured data, the blue and red lines represent the simulations from method 1 and method 2 respectively.

discrepancies that indicate limitations in the current simulation models. The primary issue is not the position of the maxima and minima, but rather the amplitude of the signal (e.g. no negative $sZ_{DR}$ is observed). Although the position of the extrema may be slightly influenced by uncertainties in mapping diameters to velocity space (see Sect. 2.1.2), the key factor affecting their position is the scattering process itself. For drops with terminal velocities up to 7 m/s, the simulations and the observations of $sZ_{DR}$ and $s\delta_{HV}$ closely align. Although, around velocities of 3.5 m/s, smaller values of $sZ_{DR}$ and bigger values of $s\delta_{HV}$



are simulated relatively to the observations. However for drops with higher terminal velocities ($v_T > 7\,ms^{-1}$), the correlation
between observations and simulated data is poor, especially for the differential reflectivity. Note these results are obtained with
perfectly oriented raindrops. When increasing the canting the amplitude of both $sZ_{DR}$ and $s\delta_{HV}$ are reduced and a worse
correlations is obtained.

## 3.2   Case Study 2: Less turbulent conditions

In this case, the notches of the Doppler spectrum are more pronounced (Fig. 9). The best-fitting gamma DSD is represented by
the blue line.

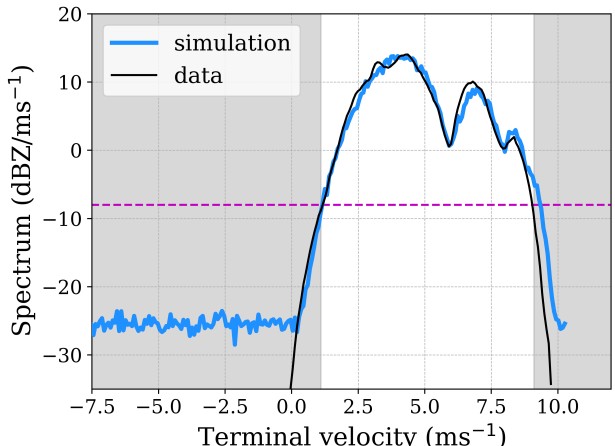

**Figure 9.** 03 February 2021, 12:40 UTC, 484m, Measured Doppler Spectrum (black line) and optimum-fitted Gamma DSD (blue line).
The purple dashed line indicates the threshold for applying the Least Squares Method in order to find the optimum fit. The parameters that
characterize the fitted spectrum are $\mu = -0.4$, $D_m = 1.6$ mm, $N_0 = 688\,mm^{-1-\mu}\,m^{-3}$, and $\sigma_t = 0.15\,ms^{-1}$

In the subsequent analysis, only one simulation method is presented, as the strong agreement between the two methods
is verified in the previous case (Sect. 3.1). In Figure 10, a comparison between simulated and observed spectral polarimetric
variables is presented. The simulations are generated using varying drop wobbling, represented by canting angle distribution
widths of 5°, 20°, and 30°. The maxima and minima for the simulated variables are found to be more pronounced relatively
to the measurements. There is sufficient agreement for the first notch of $s\delta_{HV}$ up to 5 m/s. The simulated $sZ_{DR}$ exhibits a
similar trend to the measurements; however, the amplitude of the maxima is more pronounced, and the minima are significantly
deeper. One potential cause of these discrepancies is the assumption that drops have a spheroidal shape (oblate). Therefore,
it seems plausible to conclude that the T-matrix approach using spheroids is inadequate to simulate the spectral polarimetric
variables of raindrops at higher frequencies, such as 94 GHz. The increasing canting of the drops in simulations (green, orange
and blue faint lines in Fig. 10) is causing spectral broadening, that occurs because the wobbling of the drops averages out
the distinct polarization signals over a wider range of velocities. The $sZ_{DR}$ values are spread over a wider range of Doppler





velocities, reducing the sharpness of the extrema. The more uniform distribution of drop orientations smooths out the $sZ_{DR}$ signal. Similarly, the broadening of phase differences across the spectrum leads to smeared-out minima and maxima in $s\delta_{HV}$, meaning a more gradual and continuous transition in the phase difference between horizontally and vertically polarized waves. In a nutshell, increased canting causes a more isotropic distribution of drop orientations, leading to smoother, less distinct spectral features.

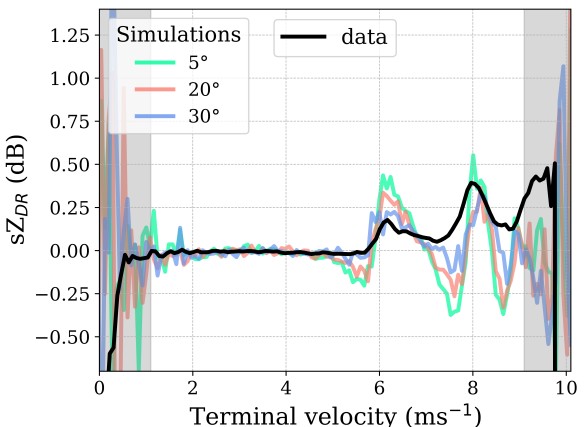
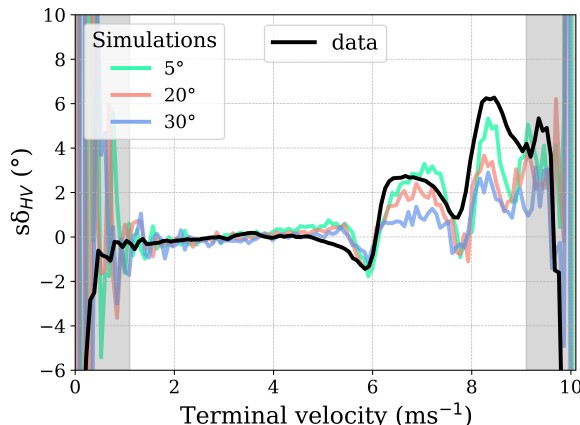

**Figure 10.** Spectral polarimetric variables of case study 2. Left panel: spectral differential reflectivity $sZ_{DR}$. Right panel: spectral differential phase $s\delta_{HV}$. The black lines represent the measured data, the green, orange and blue faint lines represent the simulations for different canting angle distribution widths (wobbling): 5°, 20°, and 30° respectively.

## 4 Conclusions and ways forward

In this study, simulations of spectral polarimetric variables were compared with real measurements in rain conditions for different levels of turbulence. The simulation accounts for factors such as the noise present in real measurements, atmospheric turbulence, and the wobbling of raindrops, aiming to replicate the complexities of actual radar data. These effects are considered to ensure a more realistic comparison between the simulated and measured spectral polarimetric variables.

The results reveal that the simulations accurately align with observations only within a limited area of the Doppler spectrum, approximately to terminal velocities up to 7 $ms^{-1}$ (i.e. equivolume diameters smaller than 2.25 mm). Overall the position of the notches in the simulations aligns well with the observations, indicating that the velocity distribution and the location of the resonances are properly captured by the simulations. However, the amplitude of the notches is not accurately represented. Notably, the simulations more accurately fit the maxima compared to the minima, especially for the differential phase. The minima in the measured data of both $sZ_{DR}$ and $s\delta_{HV}$ appear muted, while the simulated minima are significantly deeper. The maxima and minima differences are stronger in the case of lower turbulence conditions.

These discrepancies pinpoint to potential limitations in the model's treatment of the amplitude modulation caused by scattering. A potential explanation may lie in the assumption used in the T-matrix approach, which models raindrops as spheroids



or more generally as rotationally symmetric particles. However raindrops undergo oscillations (Szakáll et al., 2010), thus they may be not characterised by rotational symmetry. This suggest that traditional methods for computing scattering properties such as the well-established T-matrix method may produce inaccurate scattering parameters, especially for resonant particles (i.e. when the radar wavelength becomes comparable or smaller than the raindrop size). Other more accurate methods should

be used, e.g. the discrete dipole approximation or method of moments in the surface integral equation approach as proposed in Thurai et al. (2014); Manić et al. (2018). Future work should explore whether such more sophisticated scattering models can indeed explain the observed discrepancies. Otherwise, data acquired in low turbulence conditions can be used to build look-up tables of the polarimetric scattering properties for any given incidence angle in a data-driven approach as recently proposed by Myagkov et al. (2024).

This work paves the way toward using spectral polarimetric observations of millimeter radars for testing scattering computations of rain polarimetric variables. As such it contributes to the broader scientific community's efforts to improve cloud radar simulations, and advance our knowledge of cloud processes and their implications for atmospheric dynamics.

**Code and data availability**

The T-matrix code is available at https://github.com/jleinonen/pytmatrix. The code for the simulations that include noise and
turbulence, as well as the spectra presented in this study will be published at https://github.com/Ioanna-Tsik upon acceptance of the manuscript.

**Author contribution**

AB conceived the presented idea. AB and IT led the algorithm development, conducted the simulations and analysis. IT drafted the manuscript and designed the figures. CU led the data processing and provided methodological guidance and revisions. EM
contributed with supervision, funding acquisition, project administration and to the overall conceptual framework of the study. All authors edited and reviewed the original draft, provided critical feedback and helped shape the research, analysis and manuscript.

**Competing interests**

All authors have no competing interests.

**Acknowledgments**

This research has been supported by the PANGEA4CalVal project (Grant Agreement 101079201) funded by the European Union . We also acknowledge the support by the Hellenic Foundation for Research and Innovation (H.F.R.I.) under the



"2nd Call for H.F.R.I. Research Projects to support Post-Doctoral Researchers" (Project Acronym: REVEAL, Project Number: 07222). The work by A. Battaglia has been also supported by the European Space Agency under the activities "WInd VElocity Radar Nephoscope (WIVERN) Phase A Science and Requirements Consolidation Study" (ESA Contract Number RFP/3-18420/24/NL/IB/ab). Finally, this work used instrument and data of the Ruisdael Observatory, a scientific research infrastructure co-financed by the Netherlands Organization for Scientific Research (NWO), grant number 184.034.015.



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
