# Peer review of "Simulations of Spectral Polarimetric Variables measured in rain at W-band"

_EGUsphere, 2024_

## Referee Comment (RC2)

The manuscript under revision presents a comparison of simulated and measured spectral polarimetric variables at W-band. The analysis is done for rain assuming well known size-shape-velocity relations for raindrops. The manuscript clearly shows that more investigation is required for cloud radars to accurately simulate the spectra. The study is of a great importance for the cloud radar community. I have one major comment and several minor comments. I believe addressing these comments can considerably improve the manuscript.

Major comment:

1. Even though the Sec.2.2 is in general clear, there is a lack of explanation why it is necessary to generate noisy spectra using I/Q components. In general, average spectra can be used. These can be derived simply by adding spectral noise power to $S_{vv}$ and $S_{hh}$ (as it is done in Eq. 14 of the manuscript). Assuming no correlation between noise in the two orthogonal channels, on average there is no effect on $S_{hv}$. The variance of spectral $S_{vv}$, $S_{hh}$, and $S_{hv}$ taking into account the number of averaged spectra can be found as demonstrated in Myagkov and Ori 2022 (https://amt.copernicus.org/articles/15/1333/2022/). My question is, what are the benefits of generation of random individual spectra instead of the average ones? Please clarify this in the manuscript.

Minor comments:

1. L. 2 Change „spectral differential correlation coefficient" to "spectral correlation coefficient"
2. L. 6 "W band millimeter-wavelength radar" keep either W-band or millimeter-wavelength, these two terms are kind of redundant
3. I have a feeling that some sentences in the introduction are not well connected to each other. I recommend reformulating the text to improve the reading flow:
   a. L12-18 are about cloud radars. L.18-20 start with "Additionally" and emphasize advantages of polarimetry in precipitation radars, and then afterwards there is a jump back to cloud radars.
   b. L24-29 I understand what is meant here, but for a general reader this might be confusing. I would recommend the following sequence: Integrated variables at centimeter-wavelength are very informative. At millimeter-wavelengths signatures in integrated polarimetric variables become less pronounced. Spectral polarimetry in cloud radars is better because different particle sizes are observed independently.
   c. L29-38 I recommend making a separate paragraph and to indicate that these are some of advanced applications of polarimetric measurements at W-band.
4. The introduction section does not explain the novelty of the study, although the manuscript definitely shows novel results of comparison between state-of-the art simulations and real measurements. In the sentence (L50), there is one sentence stating that the goal is to describe the simulation. But I think the goal is much more than that,

the study shows a comparison between an advanced spectral modelling based on empirical knowledge about rain drops (including size-shape relations, size-velocity dependence, turbulence, orientation etc) and real observations. And I would put the goal of the study in the end of paragraph, i.e. after existing simulation studies have been discussed.

5. I am just curious, what is the reasoning to use a Eq.1 apparently based on studies before 2001 as a reference? And why using Thurai et al. 2008 as the second relation? Would not one be enough? Or is there a reason why two are needed, especially taking into account that the scattering simulations are often hard to distinguish in the figures?

6. Sigma on the y-axis in Fig. 2 should have VV as the subscript not just V.

7. L118-121 for me it is hard to follow these sentences. I would recommend to simply write that the broader the width of the canting angle distribution is, the lower the magnitude of the polarimetric variables.

8. Instead of Fig3 right/left I would recommend marking the panels (a) and (b) and refer to panels using these marks.

9. L124-125, elements Zij are not elements of the backscattering matrix but the Müller matrix, or as it is called in the manuscript, the phase matrix

10. L129-134 and Fig 4. Please mention that neither antenna pattern effects, nor antenna coupling for the quasi-bistatic radar configuration, nor multiple scattering, nor noise are included in the calculations of rhohv at this stage. One or a combination of these effects may drive rhohv below the stated minimum value.

11. Sec.2.1.2 again here, why using 2 parameterizations?

12. L281 Why would one expect the opposite? If I understood correctly, the same Svv,Shh, and Shv were used for both methods. The difference is only in the randomness introduced by stochastic sampling. The averaged values are expected to be the same.

13. L283 I recommend to avoid using the term correlation, when "agreement" is meant. Please check this throughout the manuscript

14. L287 L318 I see a significant difference between simulations and measurements in Fig. 8 at 5 m/s. Please check your conclusion about close alignment up to 7 m/s. Also, I do not see any noticeable differences at 3.5 m/s as written in the following sentence.

---

## Author Comment (AC1)

**REFEREE 1**

**Summary**

*Summary: This manuscript outlines simulations of spectral polarimetric radar observations for rain. The single-scattering properties are calculated using the T-matrix method for spheroids with aspect ratios determined from previous empirical relations. The spectra are then simulated using two methods for randomly generating radar signals, and the spectral ZDR and backscatter differential phase are computed from these signals and compared to observations. The simulated power spectra are fit using gamma PSD parameters and a parameter for the wind speed variability due to turbulence. The general shapes of the simulated polarimetric spectra are similar to the measurements; the magnitudes show substantial differences.*

*General comments: This manuscript provides some interesting insight into using the spectral polarimetric radar measurements to better understand rain microphysics. However, there are some issues with this study that need to be addressed before it is acceptable for publication.*

The first issue I have with this study is that the fitting of the PSD parameters and the air motion variability parameter ($\sigma_t$) are only done with respect to the spectral power. Therefore, the PSD parameters controlling the width of the particle spectrum may be compensated by the $\sigma_t$ to best fit the spectrum. As such, the assumed $\sigma_t$ may deviate substantially from the $\sigma_t$ associated with the measurements, introducing errors into the comparisons with the spectral polarimetric variables. To address this issue, the authors may want to show the sensitivity of the spectral polarimetric variables to $\sigma_t$ and the PSD parameters. Additionally, showing the range of PSD parameters and values of $\sigma_t$ that have similar RMSE during the fitting process would clarify how well constrained these parameters are.

Excluding the effects of noise and spectral averaging, the primary physical factors influencing the spectral polarimetric variables are the axis ratio–diameter relationship, the canting angle distribution, and variability in air motion, characterized by $\sigma_t$. The particle size distribution (PSD) has a comparatively minor impact on these variables. When $\sigma_t = 0$, the spectral polarimetric variables become independent of the PSD. In fitting the Doppler power spectrum, the PSD parameters that influence spectral width can be offset by variations in $\sigma_t$ to optimize the spectral fit. Consequently, rather than fitting the entire Doppler spectrum, we focus on its upper portion (above a threshold of -8 $dBZ/ms^{-1}$), which emphasizes the resonance notches—whether sharp or smoothed—providing a more robust indication of the magnitude of $\sigma_t$. This is illustrated in the following figures (1-3).

[Figure]

Figure 1: Different Do

Similarly, the authors mention that the spheroid shape may not adequately represent the scattering of natural raindrops due to processes such as drop oscillations. However, a variety of spheroids with the same fall speed but different aspect ratios could better represent this process and provide evidence as

[Figure]

Figure 2: Different mu

[Figure]

Figure 3: Different sigma

to whether the assumption of fixed aspect ratios for a given particle size is responsible for the poor comparisons between the simulated and measured spectral $Z_{DR}$. Randomly sampling aspect ratios for particles of the same fall speed would help demonstrate whether the broadening of the spectral polarimetric variables due to aspect ratio variability produces simulated spectra that are more consistent with the measurements.

We thank the reviewer for this valuable suggestion. By comparing simulated and observed spectral polarimetric variables at W-band, this study has demonstrated overall agreement, while also highlighting specific discrepancies. As such, it represents an initial step toward improving the simulation of spectral polarimetric variables at W-band. Future work will focus on investigating the potential causes of these discrepancies (this comment, next comment,....). One possible factor to explore is the assumption of a fixed aspect ratio for a given particle size.

Finally, there should be more discussion of simulating spectral polarimetric variables for radars with different transmission and reception strategies. For instance, fully polarimetric radars that transmit horizontal, receive horizontal and vertical, transmit vertical, and receive horizontal and vertical are processed differently than simultaneous transmit/receive radars. These differences could also explain some of the discrepancy between the observed and simulated spectral polarimetric variables. It is unclear what the transmission and reception strategy is for the radar observations presented in the manuscript. This information needs to be included to better understand how faithfully the method for simulating the spectral radar variables emulates the processing algorithm of the radar.

The polarimetric data acquisition will be mentioned at the beginning of section 3:
"The cloud radar measurements were obtained using a RPG Frequency Modulated Continuous Wave (FMCW) Dual Polarization W-band Cloud Doppler Radar, operating at 94 GHz in a simultaneous transmission - simultaneous reception (STSR) mode."

**Specific Comments**

• Lines 24-25: Vertically pointing radar are also able to do this. Please add that radars in slant polarization mode can take advantage of polarimetric measurements.

The text will be modified accordingly: "This configuration has the critical advantage that particles with different sizes are separated in the spectral domain (because they have different sedimentation velocities), which allows to disentangle the contributions of different particle types. While vertically

pointing radars can also achieve this separation, radars in slant polarization mode additionally exploit polarimetric measurements."

• Line 39: Do you mean vertically profiling here? Please clarify.

No, we don't mean "vertical profiling". We wrote "slant or horizontal profiling" because meaningful polarimetric variables are obtained at low/intermediate elevation angles. However, there is an exception for the linear depolarization ratio, which can be used for applications involving vertical profiling.

• Lines 50-52: "Describing the methodology to compute spectral polarimetric variables" doesn't really address a science question. Based on the previous line and my impression of the study, a stronger goal might be to explore how different assumptions impact the simulated spectral polarimetric variables.

The text will be modified to "Goal of this study is to explore how different assumptions that are related to atmospheric conditions (turbulence) and white and stochastic noise of a real radar spectrum, impact the simulated spectral polarimetric variables.The second objective is to present a novel comparison between simulated and observed data."
This text will be placed at the end of the introduction before the description of the paper structure.

• Lines 57-58: The T-matrix method can simulate the scattering properties of arbitrary shaped particles (as long as the numerical integration converges; Wriedt 2002). However, these codes are not widely available and are much less efficient. Please change this sentence accordingly.

The text will be modified to "However, raindrops generally change due to oscillations, which cause departure from rotationally symmetric shape. The T-matrix method can, in principle, simulate scattering from non-rotationally symmetric particles (given numerical convergence; Wriedt 2002), but such implementations are computationally demanding and not widely available. As a result, most T-matrix applications rely on the assumption of rotationally symmetric particles."

• Lines 74-75: Does equation (1) come from one of these studies specifically? Please clarify.

This will be specified.
"In the following, Eq. (1) is employed to describe the raindrop axis ratio. For small raindrops, the axis ratio follows the parameterization by Keenan et al. (2001), while for larger drops, the formulation of Beard and Chuang (1987) is applied."

• Line 144: Please add some reference(s) for these equations.

We have cited Altas et al., (1973) where the relationship between drop diameters and corresponding terminal velocities is described, and the relevant curves are shown in Figure 5.

• Line 157: Where does this equation come from? Please address.

A reference will be added.

• Line 158: Shouldn't this denominator be in terms of d $v_los$ instead of d $v_t$? What if the horizontal wind is large compared to the fall velocities or the radar has an elevation angle near 0? If this equation is an approximation, please indicate under what conditions it is valid.

For clarification, we will add the following sentence:
"Eq. (8) is formulated for elevation angles $\theta_{el}$ significantly greater than zero, without accounting for the contribution of turbulence."

• Line 178: It is a bit unclear why these two methods for calculating the spectra are being discussed. Are they the only two such methods? Please add some brief discussion on this point at the end of the previous section.

A section 2.2.5, Rationale for simulation based on IQ, will be added in the revised manuscript.

• Lines 240-241: What is the transmission and reception strategy of this radar? Can it measure retrieved co-polar signals independently?

We will mention in the text that the radar operates in a simultaneous transmission - simultaneous reception (STSR) mode.

• Line 257: I think the "e" is missing from the subscript on the elevation angle symbol.

We will change to $\theta_{el}$.

• Line 264: Please be more specific about the degree of turbulence during this case and the following case. Are these cases on the extreme ends of the turbulence that might be observed during these such events?

No, these cases are not on the extreme ends of the observed turbulence. Therefore, the titles of the subsections will be changed:
3.1 Case Study 1: Moderate turbulence conditions
3.2 Case Study 2: Light turbulence conditions

• Lines 278-279: How were they adjusted? According to the measured values of the smallest particles? Please clarify.

Yes, the corrections were determined by measuring the offset observed for the smallest particles, which are assumed to be spherical. The $sZ_{DR}$ and $s\delta_{HV}$ values were adjusted such that the smallest particle measurements align with 0 dB and 0°, respectively.

• 7: What does the gray shading on the plot represent? Please add this information to the figure caption.

We will add the following sentence in the text relating to Fig. 8:
"The spectral polarimetric variables are analyzed outside the gray-shaded regions, where the Doppler spectral power exceeds -8 $dBZ/ms^{-1}$, to ensure a sufficiently high signal-to-noise ratio."

---

## Author Comment (AC2)

**REFEREE 2 - Alexander Myagkov**

**Summary**

The manuscript under revision presents a comparison of simulated and measured spectral polarimetric variables at W-band. The analysis is done for rain assuming well known sizeshape-velocity relations for raindrops. The manuscript clearly shows that more investigation is required for cloud radars to accurately simulate the spectra. The study is of a great importance for the cloud radar community. I have one major comment and several minor comments. I believe addressing these comments can considerably improve the manuscript.

**Major comment**

Major comment: 1. Even though the Sec.2.2 is in general clear, there is a lack of explanation why it is necessary to generate noisy spectra using I/Q components. In general, average spectra can be used. These can be derived simply by adding spectral noise power to Svv and Shh (as it is done in Eq. 14 of the manuscript). Assuming no correlation between noise in the two orthogonal channels, on average there is no effect on Shv. The variance of spectral Svv, Shh, and Shv taking into account the number of averaged spectra can be found as demonstrated in Myagkov and Ori 2022 (https://amt.copernicus.org/articles/15/1333/2022/). My question is, what are the benefits of generation of random individual spectra instead of the average ones? Please clarify this in the manuscript.

The reason we chose to generate noisy spectra using I/Q components, instead of working with average spectra with added noise power, is to explicitly investigate whether the use of random individual noisy spectra can help explain or reproduce the variability and degradation often observed in measured spectral polarimetric variables, particularly in variables that rely on cross-channel correlations like $S_{hv}$, at low SNR and low correlations where approximated formulas as proposed in Myagkov and Ori 2022 (https://amt.copernicus.org/articles/15/1333/2022/) tend to fail. We will include such considerations in the revised version.

By simulating the noisy spectra from I/Q components, we aimed to test whether noise characteristics contribute to the spectral variability seen in observations and whether this could help explain the persistent discrepancies between simulated and observed spectral polarimetric variables. In this sense, our work seeks to fill a gap in the literature and offer an alternative angle to understanding the role of noise in radar polarimetry.

**Minor comment**

1. L. 2 Change "spectral differential correlation coefficient" to "spectral correlation coefficient"

Will be done in the revised manuscript.

2. L. 6 "W band millimeter-wavelength radar" keep either W-band or millimeterwavelength, these two terms are kind of redundant

Will be done in the revised manuscript.

3. I have a feeling that some sentences in the introduction are not well connected to each other. I recommend reformulating the text to improve the reading flow:
a. L12-18 are about cloud radars. L.18-20 start with "Additionally" and emphasize advantages of polarimetry in precipitation radars, and then afterwards there is a jump back to cloud radars.

Rephrasing will be performed for improving the reading flow.

b. L24-29 I understand what is meant here, but for a general reader this might be confusing. I would recommend the following sequence: Integrated variables at centimeter-wavelength are very informative. At millimeter-wavelengths signatures in integrated polarimetric variables become less pronounced. Spectral polarimetry in cloud radars is better because different particle sizes are observed independently.

This part will be rewritten.

*"This configuration has the critical advantage that particles with different sizes are separated in the spectral domain (because they have different sedimentation velocities), which allows to disentangle the contributions of different particle types. While vertically pointing radars can also achieve this separation, radars in slant polarization mode additionally exploit polarimetric measurements. At higher frequencies, where multiple resonances occur across the particle size distribution (PSD), the polarimetric variables—resulting from integration over the entire PSD—tend to average out the characteristic features of single-particle scattering, often balancing positive and negative contributions. Consequently, these variables exhibit low sensitivity to PSD variations. Further, they reflect both scattering and propagation effects. A way to mitigate these challenges at millimeter wavelengths is to analyze polarimetric variables in the spectral domain."*

c. L29-38 I recommend making a separate paragraph and to indicate that these are some of advanced applications of polarimetric measurements at W-band.

A separate paragraph will be made.

4. The introduction section does not explain the novelty of the study, although the manuscript definitely shows novel results of comparison between state-of-the art simulations and real measurements. In the sentence (L50), there is one sentence stating that the goal is to describe the simulation. But I think the goal is much more than that, the study shows a comparison between an advanced spectral modelling based on empirical knowledge about rain drops (including size-shape relations, size-velocity dependence, turbulence, orientation etc) and real observations. And I would put the goal of the study in the end of paragraph, i.e. after existing simulation studies have been discussed.

The rephrased goals of the study will be placed at the end of the paragraph as recommended by the reviewer.

*"Therefore, the first goal of this study is to explore how different assumptions that are related to atmospheric conditions (turbulence) and white and stochastic noise of a real radar spectrum, impact the simulated spectral polarimetric variables. The second objective is to present a novel comparison between simulated and observed data."*

5. I am just curious, what is the reasoning to use a Eq.1 apparently based on studies before 2001 as a reference? And why using Thurai et al. 2008 as the second relation? Would not one be enough? Or is there a reason why two are needed, especially taking into account that the scattering simulations are often hard to distinguish in the figures?

We used two axis ratio parameterizations to demonstrate that the choice of parameterization does not have a significant impact on the polarimetric variables.

The first relation (Eq.1) was included as it has been traditionally used in scattering simulations, while the Thurai et al. (2008) relation was included because it is based on more recent measurements and is widely adopted in polarimetric studies.

To further illustrate the limited influence of the axis ratio parameterization, we plotted the polarimetric variables in Figures 3 and 4, showing that the differences between the two parameterizations are minimal.

6. Sigma on the y-axis in Fig. 2 should have VV as the subscript not just V.

Will be done in the revised manuscript.

7. L118-121 for me it is hard to follow these sentences. I would recommend to simply write that the broader the width of the canting angle distribution is, the lower the magnitude of the polarimetric variables.

This will be rephrased.

*"In Fig. 3 (right), $\delta_{HV}$ remains near zero for small drop diameters, consistent with Rayleigh scattering. As the diameter increases, $\delta_{HV}$ departs from zero and exhibits oscillatory behavior, attributed to resonance effects and the transition from spherical to oblate shapes. These fluctuations become more pronounced at larger diameters. Variability in drop orientation within the radar sampling volume, described*

*by canting angle distributions, further contributes to the observed variations in $\delta_{HV}$. The broader the width of the canting angle distribution is, the lower the magnitude of the polarimetric variables."*

8. Instead of Fig3 right/left I would recommend marking the panels (a) and (b) and refer to panels using these marks.

We thank the reviewer for this recommendation, it is really appreciated. However, in order to maintain consistency with the style used throughout the manuscript (and in similar figures), we have chosen to continue referring to the panels as "left" and "right."

9. L124-125, elements Zij are not elements of the backscattering matrix but the Müller matrix, or as it is called in the manuscript, the phase matrix

Will be corrected.

10. L129-134 and Fig 4. Please mention that neither antenna pattern effects, nor antenna coupling for the quasi-bistatic radar configuration, nor multiple scattering, nor noise are included in the calculations of rhohv at this stage. One or a combination of these effects may drive rhohv below the stated minimum value.

Thank you for this clarification, which is will be mentioned in the article.

11. Sec.2.1.2 again here, why using 2 parameterizations?

As explained previously, we used two parameterizations to demonstrate that the choice of raindrop fall velocity model does not significantly impact the simulated polarimetric variables. This was important to show the robustness of the results against different commonly used relations.

12. L281 Why would one expect the opposite? If I understood correctly, the same Svv,Shh, and Shv were used for both methods. The difference is only in the randomness introduced by stochastic sampling. The averaged values are expected to be the same.

Yes indeed agreement is expected. We will rephrase the statement. The use of both methods was to ensure that the stochastic perturbations respect the physical relationships between the scattering elements. The fact that both methods demonstrated consistency when producing the polarimetric variables provided confidence in the turbulence generation on the simulations and that the discrepancies in the observations were not due to a simulation artefact.

13. L283 I recommend to avoid using the term correlation, when "agreement" is meant. Please check this throughout the manuscript

We will avoid the term correlation when discussing the results.

14. L287 L318 I see a significant difference between simulations and measurements in Fig. 8 at 5 m/s. Please check your conclusion about close alignment up to 7 m/s. Also, I do not see any noticeable differences at 3.5 m/s as written in the following sentence "For drops with terminal velocities up to 7 m/s, the simulations and the observations of $sZ_{DR}$ and $s\delta_{HV}$ show reasonable agreement. Although, around velocities of 5 m/s, smaller values of $sZ_{DR}$ and bigger values of $s\delta_{HV}$ are simulated relatively to the observations."

In section 4 - conclusions and ways forward, we will modify:
"The results reveal that the simulations closely align and show reasonable agreement with observations only within a limited area of the Doppler spectrum, approximately to terminal velocities up to 5 and 7 $ms^{-1}$ (i.e. equi-volume diameters smaller than 1.33 and 2.25 mm), respectively."

---

## Author Response (AR2)

**We sincerely thank the reviewers for their valuable comments and suggestions, which have helped improve the quality of the paper.**

----- For clarity: Reviewer comments are shown in black, Author responses are shown in RED. -----

**-------------------- *REFEREE 1* --------------------**

**Summary**: The authors have done a good job addressing most of my prior comments in this version of the manuscript. However, I have some additional minor comments that should be addressed before the manuscript is accepted for publication.

**General comments**:

- Lines 28-29: Please add a reference for this statement.

We have cited the reference by Kollias et al. (2011)
Kollias, P., Remillard, J., Luke, E., and Szyrmer, W.: Cloud radar Doppler spectra in drizzling stratiform clouds: 1. Forward modeling and remote sensing applications, Journal of Geophysical Research: Atmospheres, 116, 2011. DOI:10.1029/2010JD015238.

- Line 30: Please mention which variables are minimally impacted by the PSD.

We have changed to "*This is especially evident in the simulations of differential reflectivity ($Z_{DR}$) where this parameter exhibits very low values and sensitivity to PSD variations (Unal and van den Brule, 2024).*"

- Line 67: Please clarify this statement since white noise is by its nature stochastic.

The sentence is modified to: "*Therefore, the first goal of this study is to explore how different assumptions that are related to atmospheric conditions (turbulence) and white noise of a real radar spectrum, impact the simulated spectral polarimetric variables.*"

- Lines 81-82: Please explain why two different raindrop aspect ratio parametrizations are used in this study. Also, please clarify that the second equation written above is from Andsager et al. (1999) and is their fit to the Beard and Chuang (1987) model.

The use of two different axis ratio parameterizations reflects the fact that small and large raindrops deform differently due to competing physical forces. For small drops, surface tension dominates, and they remain nearly spherical. The Keenan et al. (2001) parameterization is well-suited to describe this regime. For larger drops, aerodynamic forces become more significant, leading to more oblate shapes. The Beard and Chuang (1987) better captures this deformation behavior. We changed the text to: "*... where a/b denotes the ratio of the major to minor axis of the oblate spheroid. The use of two different formulations reflects the physical differences in raindrop deformation regimes. For small raindrops, the axis ratio follows the parameterization by*

*Keenan et al. (2001), while for larger drops, the fit of Andsager et al. (1999) to the Beard and Chuang (1987) model is used.*"

- Line 90: This is the relative permittivity, not the refractive index. Please fix.

The sentence is modified to: *"The raindrops are assumed to be at 10 degrees C and the complex relative permittivity of water at this temperature is 3.2 -1.8j at 94 GHz (Lhermitte, 1990)"*

- Line 121: Please add that backscatter differential phase also depends on size.

The text is modified to *"The differential phase ($\delta_{HV}$) refers to the phase shift introduced at backscattering between the horizontally and vertically polarized components of the received radar signal. This parameter depends on the size of the hydrometeors and provides information about their shape and orientation."*

- Line 137: Please clarify here that a distribution of orientations is needed to reduce RhoHV; a uniform non-zero orientation still gives RhoHV=1.

We have clarified this point in the revised text: *"On the other hand, raindrops with variations in orientation or tilt of the drop axis relatively to the direction of motion (canting) have $\rho_{HV}$ slightly lower than 1, showing a minimum loss of correlation between the two different polarization states."*

- Line 162: Please provide some brief motivation here for why these two methods are presented.

The use of both methods was to ensure that the stochastic perturbations respect the physical relationships between the scattering elements. The fact that both methods demonstrated consistency when producing the polarimetric variables provided confidence in the turbulence generation on the simulations and that the discrepancies in the observations were not due to a simulation artifact. The following is added: *"The use of both approaches ensures that the introduced stochastic perturbations respect the physical relationships between scattering elements. Their agreement increases confidence in the simulated turbulence structure and supports that observed discrepancies are not artifacts of the simulation method."*

- Line 268: This relation assumes that significant shear layers are not present at higher altitudes.

We agree that wind shear can generate mechanical turbulence, particularly in regions with strong wind gradients, such as near jet streams or inversions. However, in the context of our study, turbulence at lower levels, which is primarily driven by surface heating and friction, dominates the signal variability and uncertainty. At higher altitudes, shear layers may exist, but they generally contribute less to the turbulence detected by our system compared to the intense mixing closer

to the surface. It is also possible to have higher turbulence at high heights in a convective precipitation system (for example in thunderstorms with large updrafts). Therefore, the sentence *"Higher altitudes are usually characterized by significantly less turbulence relatively to lower levels, because turbulence is often generated by surface heating and friction."* is meant in a relative sense. We modified as following to clarify the turbulence origin: *"Excluding cases of strong wind shear (e.g., jet streams) and deep convective systems (e.g., thunderstorms), higher altitudes are generally characterized by significantly less turbulence than lower levels, as turbulence is mostly generated by surface heating and friction."*

- Line 298: Please clarify if these results are for the PSD parameters fit to the Doppler spectrum in Fig. 7.

The sentence is modified to: *"In Figure 8, the black lines represent the measured spectral polarimetric variables $sZ_{DR}$ (left) and $s\delta_{HV}$ (right), while the blue and red lines are the results of the two simulation methods, by using the above mentioned optimum-fitted Doppler spectrum (see Fig. 7)."*

- Line 302: Maybe I'm missing something, but how is the dependence of the spectral polarimetric variables on the PSD tested if a single PSD is derived from fitting the Doppler spectrum to the data?

This part is modified to: *"Next to the radar elevation angle, the primary physical factors influencing the spectral polarimetric variables are the axis ratio–diameter relationship, the canting angle distribution (Unal and van den Brule, 2024), and variability in air motion, characterized by $\sigma_t$. The values of $sZ_{DR}$ and $s\delta_{HV}$ do not depend on the raindrop size distribution (Unal and van den Brule, 2024). However, what may vary in Figure 8 is the terminal velocity range—for example, under low turbulence conditions, the velocity range narrows when $D_m$ is small, as in the case of light rain."*

In the following figure, we have computed $sZ_{DR}$ and $s\delta_{HV}$ with different PSDs keeping the same turbulence, canting angle distribution and axis ratio-diameter.

[Figure]

*Figure 1 Doppler spectrum, spectral $Z_{DR}$ and spectral $\delta_{HV}$, computed with the same turbulence ($\sigma_t$ =0.1m/s), canting angle distribution and axis ratio-diameter relation, but with **different PSDs**.*

----- For clarity: Reviewer comments are shown in black, Author responses are shown in BLUE. -----

**---------------- *REFEREE 2* (Alexander Myagkov) ----------------**

**Summary**: I would like to thank the authors for thoroughly considering my suggestions to improve the manuscript. All of my concerns have been properly and comprehensively addressed.

I kindly request one final wording correction: in line 262 (sec. 2.2.5) of the revised manuscript, please replace the word "proposed" with "demonstrated". This better reflects the aim of the cited publication, which is to show that widely accepted polarimetric variables complicate accurate estimation of the effects of random errors.

After this correction, I fully recommend the manuscript for publication.

We have changed from "proposed" to "demonstrated".